# Experimental uninephrectomy associates with less parasympathetic modulation of heart rate and facilitates sodium-dependent arterial hypertension

**Rainer U. Pliquett** [1,2]*, **Ralf P. Brandes**[1]

**1** Institute of Cardiovascular Physiology, Vascular Research Centre, Fachbereich Medizin, Goethe University, Frankfurt (Main), Germany, **2** Department of Nephrology & Diabetology, Carl-Thiem Hospital, Cottbus, Germany

* rpliquett@endothel.de

## Abstract

### Background

Blood pressure is known to be increased in kidney donors following living-donor kidney transplantation. However, the physiological underpinnings of the blood-pressure increase following uninephrectomy remain unclear. We hypothesized that changes in sympathetic tone or in parasympathetic modulation of sinus node function are involved in the blood-pressure increase following experimental kidney-mass reduction.

### Methods

C57BL6N mice (6 to 11 per group) subjected to sham surgery (controls) or uninephrectomy with or without a one-week course of sodium chloride-enriched, taurine-deficient diet were studied. Uninephrectomized mice treated with a subcutaneous infusion of angiotensin-II over a period of one week were positive controls. A transfemoral aortic catheter with telemetry unit was implanted, readings of heart-rate and blood-pressure were recorded. Power-spectral analysis of heart rate and systolic blood pressure was performed to gain surrogate parameters of sympathetictone and parasympathetic modulation of sinus node function. Baroreflex sensitivity of heart rate was determined from awake, unrestrained mice using spontaneous baroreflex gain technique.

### Results

Systolic arterial blood pressure, heart rate and baroreflex sensitivity were not different in uninephrectomized mice when compared to controls. Parasympathetic modulation of sinus node function was less in uninephrectomized mice in comparison to controls. Uninephrectomized mice of the high-angiotensin-II model or of the high-salt and taurine-deficiency model had an increased systolic arterial blood pressure.

**Data Availability Statement:** All relevant data are within the paper and Supporting Information files.

**Funding:** RUP received grant from "Deutsche Nierenstiftung" 2008 (https://www.nierenstiftung.

de). The funders had no role in study design, data collection and analysis, decision to publish, or preparation of the manuscript.

**Competing interests:** The authors have declared that no competing interests exist.

## Conclusions

Uninephrectomy associated with less parasympathetic modulation of sinus node function. The combination of uninephrectomy, taurine-deficiency and high-salt intake led to arterial hypertension.

## Introduction

Fifty percent of children born with a single kidney developed arterial hypertension by the age of 18 years, 40% of them needed a renal replacement therapy by the age of 30 years [1]. Following living-donor kidney transplantation, donors were shown to have a propensity for arterial hypertension [2–4]. Specifically, hypertension developed in 4%, 10%, and 51% of kidney donors at 5, 10, and 40 years post-transplant, respectively [2], and the development of hypertension among kidney donors prevailed after correction for aging [3]. In addition, female kidney donors were shown to have more gestational hypertensive complications than pregnant women from a matched non-donor cohort [5]. In non-dialysis patients following uninephrectomy, 24-hour blood-pressure recordings showed a higher rate of a non-dipper pattern [6]. To address the increasing demand for kidney transplants, safety data are needed with regard to donor outcome.

In an animal model of subtotal (5/6) nephrectomy, systolic blood pressure was shown to be increased [7]. Following uninephrectomy, a cell-cycle dependent, amino-acid mediated and target of rapamycin—dependent hypertrophy of the remaining kidney was found [8, 9]. The responsiveness of renal afferent nerves was shown to be enhanced by high urinary sodium concentrations [10]. However, in mice, the efferent renal sympathetic nerve activity was not affected by a high-salt diet alone [11].

In the present study, the influence of uninephrectomy both on blood-pressure and autonomic-nervous-system regulation was investigated. In addition to uninephrectomy, the role of a hypertensive challenge on autonomic nervous system function and on blood-pressure regulation was investigated. As a hypertensive challenge, a high-salt, low-taurine diet was applied. Besides a salt-rich diet, the amino acid L-alanine was added to reduce cellular taurine uptake via the taurine transporter to induce intracellular taurine deficiency, thus counteracting the antihypertensive effect of taurine [12].

Here we hypothesized, firstly, that blood pressure is higher in uninephrectomized mice than in controls, and secondly, that uninephrectomy leads to changes in sympathetic tone and/or in parasympathetic modulation of sinus node function when compared with controls. In addition, we hypothesized that salt overload with taurine deficiency increases systolic blood pressure in uninephrectomized mice to a comparable extent as in hypertensive mice on a standard dose of subcutaneous angiotensin-2 stimulation.

## Methods

### Animal models

Male C57BL/6N mice (6–15 per group, age: 10–12 weeks, Charles River, Sulzfeld, Germany) were housed in individual cages in a separate room under standard conditions (21˚C, 12 h dark-light cycle), standard chow and drinking water ad libitum. Daily care was provided at the same time, body weight was taken weekly. All animal procedures and experiments adhered to the APS's Guiding Principles in the care and use of vertebrate animals in research and training,

# Flow chart of Interventions

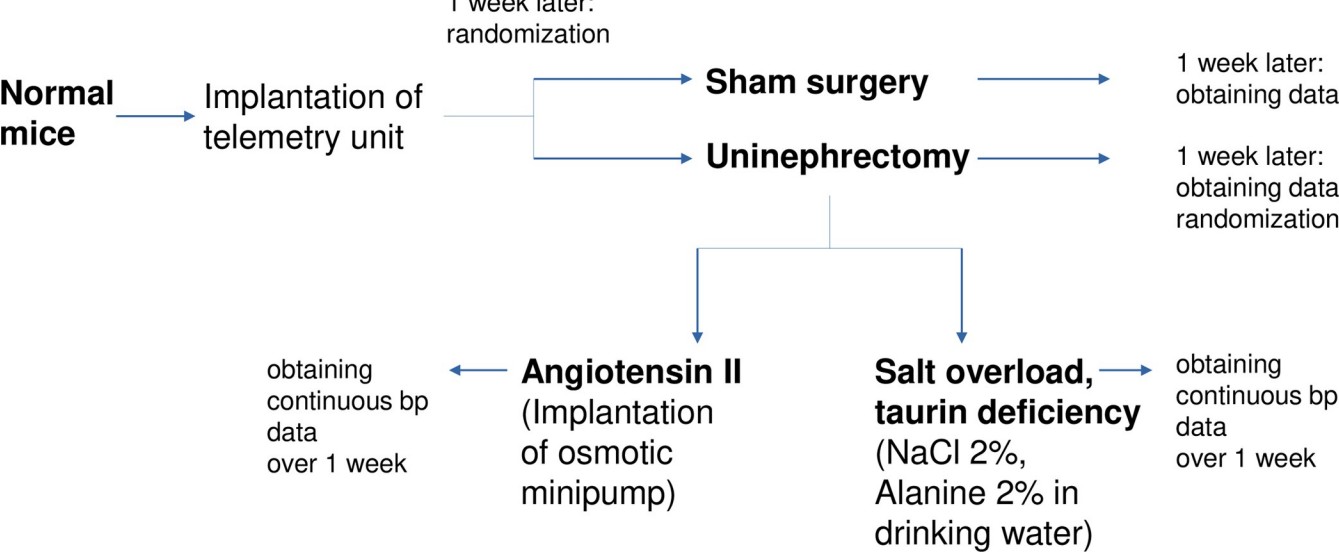

**Fig 1. Experimental setting of murine models uninephrectomy and secondary arterial hypertension.**

all possible steps were taken to avoid animal suffering at each stage of the experiments. Ethical approval was obtained from local animal-care officials and the supervising federal authority (approval number: V54-19c20/15F28K2154 issued by Regierungspräsidium Darmstadt, Hessen, Germany). The experimental setting was displayed in Fig 1. In all surgeries, an inhalational anaesthesia using isoflurane (2% initially, 0.8–1% continuously) and subcutaneous fentanyl (0.06 mg/kg) were used. After surgery, subcutaneous buprenorphine (0.3 mg/kg) was applied for pain relief and, as empiric antibiotic prophylaxis, subcutaneous ampicillin (50 mg/kg) was administered. Transfemoral aortic catheters with telemetry units (TA11PAC10, Data Sciences International, St. Paul, Minnesota, USA) were implanted in all animals as described previously [13].

One week later, one group of mice was subjected to a sham surgery (n = 10) which consisted of skin incision plus manipulation of one kidney. Another group was subjected to uninephrectomy (n = 15). After one week of recovery, hemodynamic parameters were obtained and powerspectral analysis performed as outlined below. Afterwards, as an add-on study, uninephrectomized mice were randomly assigned to a model of hypertension due to salt overload and taurine deficiency (n = 6) or to an established angiotensin-II-dependent hypertension model as a positive control group (n = 6). Angiotensin-II-dependent hypertension was achieved by a subcutaneous implantation of Alzet osmotic minipumps (Cupertino, California, USA) filled with sterile [5val]-angiotensin-II solution. The osmotic minipumps were administered for a maximum duration of 14 days. The daily dose released into the body was 1.4 mg per kg body weight. Animals subjected to salt overload and taurine deficiency only had access to drinking water with added sodium-chloride (2%) and L-alanine (2%) over one week as published previously for uninephrectomized rats [14]. In the present study, this model of hypertension was adapted to mice. For analysis, a within-group comparison of daily mean and peak systolic arterial pressure at baseline and at day 7 was performed. After finishing the experiments, the mice were sacrificed by decapitation under isoflurane anaesthesia.

## Analysis of blood pressure, heart rate, powerspectrum, baroreflex sensitivity

After one week of recovery from sham surgery or uninephrectomy and during the one-week period of secondary hypertension induction, telemetric blood-pressure readings of individual animals were obtained every 5 minutes for 10 seconds until the animals were sacrificed. Pulse intervals (defined as consecutive dP/dtmax) were extracted from aortic blood pressure wave-forms using ART 4.2 Gold software (Data Sciences International; St. Paul, Minnesota, USA). Mean heart rate, mean and peak systolic and diastolic blood pressure were determined over 24 hours. In addition, after recovery following sham surgery or uninephrectomy and 1 week after onset of secondary hypertension, a one-hour continuous baseline recording was taken in the morning in unrestrained, resting conditions. The last 30 minutes were used for powerspectral analysis and for determination of spontaneous baroreflex sensitivity as reported previously [13]. The telemetric aortic blood pressure readings were transmitted from the implanted telemetry unit to a receiver placed below the mouse cage, digitized with a sampling rate of 1000 Hz. After linear interpolation with an equidistant sampling interval of 0.05 s (20 Hz), power spectral analyses of systolic blood pressure and of pulse intervals were performed using Fourier transformation (1024-point series corresponding to a 51.2-s period). Each spectral band obtained was a harmonic of 20/1024 Hz (0.019 Hz). The power spectral analysis of both systolic blood pressure and pulse intervals yielded intensities (units: $mmHg^2$ and $ms^2$) using the low-frequency band (0.15–0.6 Hz) for power spectrum of systolic blood pressure (LF-SBP) and the high-frequency band (2.5–5.0 Hz) for power spectrum of heart rate (HF-HR). Cumulative intensity of LF-SBP was considered as quantitative measure of sympathetic modulation of vascular tone, whereas the cumulative intensity of HF-HR was regarded as a quantitative measure of parasympathetic modulation of sinus node function [15, 16]. Baroreflex sensitivity of heart rate was determined by spontaneous baroreflex gain technique. Sequences of concomitant spontaneous changes of systolic blood pressure (of at least 15 mmHg) and pulse intervals of at least 4 consecutive heart beats were digitized, linearly interpolated and correlated utilizing the Hemolab software [17, 18]. For individual ramps in blood pressure and pulse intervals, a correlation coefficient of at least 0.9 was set as threshold for data inclusion for the analysis. A time delay of 0 seconds was chosen for analysis of concomitant blood-pressure and pulse-interval changes. The average of at least 10 individual baroreflex ramps (linear portion of systolic blood pressure–pulse-interval relationship) was considered as baroreflex sensitivity of heart rate.

## Statistics

Results are given as means ± standard deviation. For inter-group comparisons with equal variances, one-way ANOVA / Tukey's multiple comparison post-hoc test or unpaired two-tailed student's t-test were used, where appropriate. For comparisons within groups, paired Student's t test was used. If the normality test failed, nonparametric tests (Kruskal Wallis test / Dunn's post-hoc test or–for two-groups—Mann-Whitney-U or Wilcoxon-matched pairs test) were used, where appropriate. Asterisks highlight different p values: *p<0.05; **p<0.01; ***p<0.001. Statistical analysis was carried out with Graphpad (La Jolla, California, USA).

## Results

### Hemodynamic characteristics of unilaterally nephrectomized mice

Heart rate, mean and peak systolic blood pressure, diastolic and pulse pressure did not differ between normal controls and uninephrectomized mice (Fig 2). However, peak systolic arterial

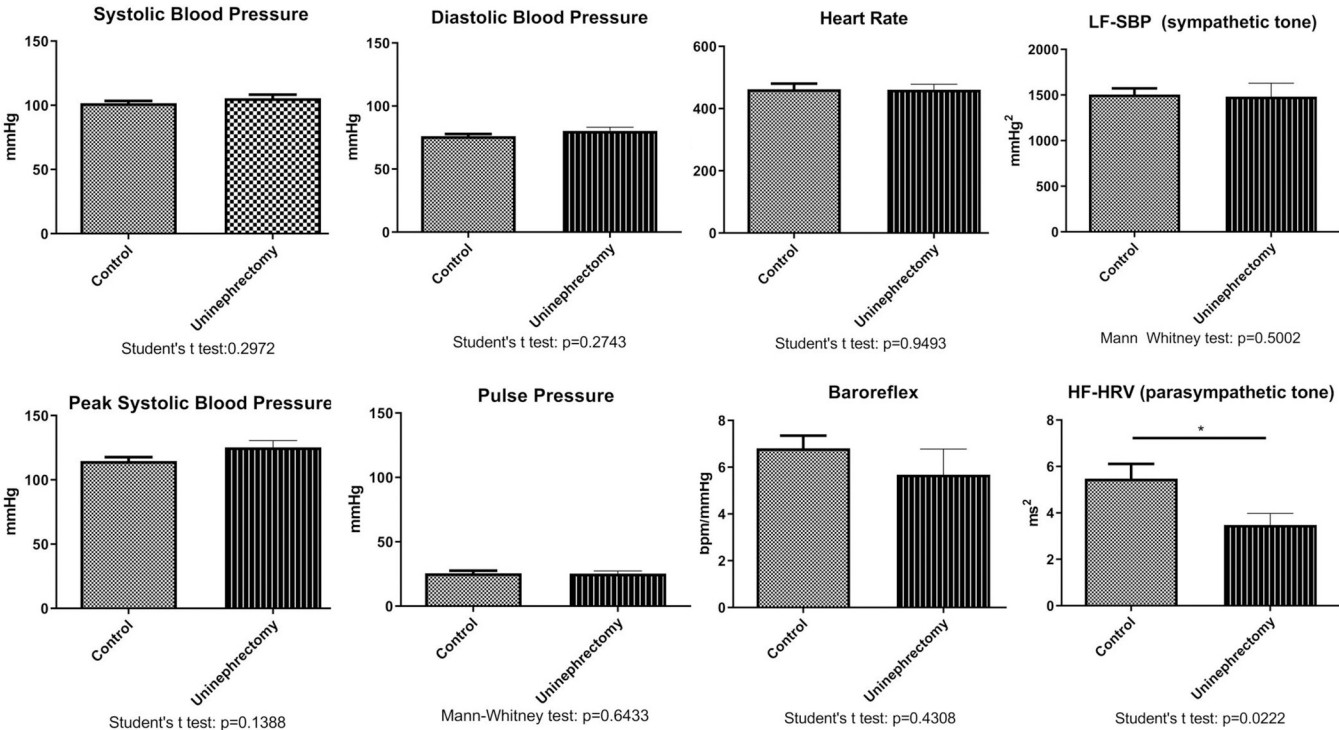

**Fig 2. Hemodynamic data (mean and peak systolic, diastolic, pulse pressure, heart rate) and powerspectral-analysis data low-frequency band of powerspectrum of systolic blood pressure (LF-SBP) as a measure of sympathetic modulation of vascular tone, high-frequency band of powerspectrum of heart rate (HF-HR) as a measure of parasympathetic modulation of sinus node function and baroreflex-sensitivity data of sham-operated C57BL/6n mice (controls, n = 8) and unilaterally nephrectomized C57BL/6n mice (n = 15).**

blood pressure tended to be higher in uninephrectomized mice (125.3 ± 20.3 mmHg) as compared to controls (114.7 ± 9.5 mmHg), although this difference did not reach the significance level. Likewise, although mean heart rate was not different (uninephrectomized mice: 460.7 bpm, normal controls: 462.4 bpm), uninephrectomized mice reached a lower minimal (336.5 bpm) and maximal (563.9 bpm) heart rate than controls (minimal heart rate:403.4 bpm, maximal heart rate: 567.6 bpm) under resting conditions.

## Parasympathetic modulation of sinus node function is reduced in uninephrectomized mice

In comparison to controls, sympathetic modulation was similar in uninephrectomized mice (Fig 2). However, parasympathetic modulation of sinus node function was reduced in uninephrectomized mice. In uninephrectomized mice, baroreflex sensitivity of heart rate of did not differ between groups. However, in uninephrectomized mice, a non-significant tendency to a reduced baroreflex sensitivity of heart rate was demonstrated (5.7 ± 3.6 bpm/mmHg versus 6.8 ± 1.6 bpm/mmHg).

## Uninephrectomized mice with sodium-chloride overload and taurine deficiency show a predisposition for hypertension

When compared to baseline, uninephrectomized mice subjected to a high-salt and a low-taurine intake had a higher mean systolic arterial pressure (Fig 3) and a higher peak arterial pressure (Fig 4) after one week. Likewise, hypertensive mice subjected to a standard dose

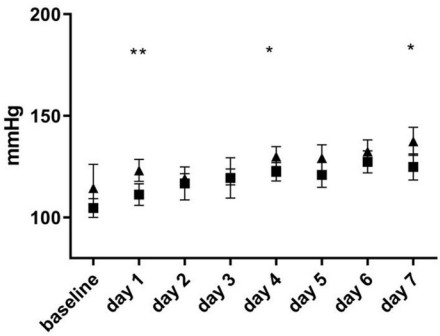

**Mean systolic blood pressure in hypertension model of salt overload and taurin deficiency versus angiotensin II**

■ uninephrectomy, 2%NaCl, 2%ALA (drinking water)

▲ uninephrectomy, angiotensin II (SC)

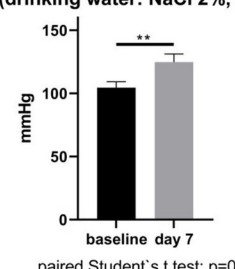

**Daily mean systolic arterial pressure in uninephrectomized mice on high-salt diet, taurin deficiency (drinking water: NaCl 2%, L-Ala 2%)**

paired Student`s t test: p=0.005

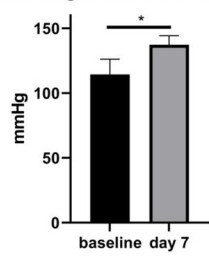

**Daily mean systolic arterial pressure in uninephrectomized mice on s.c. Angiotensin-II stimulation**

paired Student`s t test: p=0.049

**Fig 3. Upper panel: evolution of daily mean systolic blood pressure in uninephrectomized mice during hypertension-inducing interventions.** a) oral sodium-chloride load and taurine deficiency (NaCl (2%) + L- alanine (2%)-enriched drinking water); b) angiotensin II: 1.4 mg/kg/d subcutaneously applied by osmotic minipump. Lower panel: direct comparison of daily mean systolic blood pressure within groups (baseline versus end of study).

angiotensin-II treatment had a higher mean systolic and peak systolic after one week, when compared to baseline. Specifically, during the week of hypertension development, daily mean systolic arterial blood pressure was similar in uninephrectomized mice of both groups on 4 of 7 days under investigation. However, on 5 of 7 days, daily peak systolic blood pressure was higher in the positive control group on angiotensin-II treatment than in the group on salt-rich, taurine-deficient diet. As for the autonomic nervous system, parasympathetic modulation of heart rate (HF-HR) was not different between both hypertensive groups, however, sympathetic modulation of vascular tone (LF-SBP) was higher in the high-angiotensin-II model $(2014 \pm 128.0 \text{ mmHg}^2 \text{ versus } 1604 \pm 87.5 \text{ mmHg}^2, p = 0.0038)$.

## Discussion

While the high–angiotensin II model associated with sympathoactivation, uninephrectomized mice showed less parasympathetic modulation of sinus node function as compared to controls without a reduced kidney mass. In addition, when salt overload and taurine deficiency were applied to uninephrectomized mice, arterial hypertension developed within the same time frame as in the angiotensin-II model of hypertension. The underlying mechanism causing this difference in parasympathetic modulation of sinus node function in uninephrectomized animals versus normal controls remains unclear. In uninephrectomized animals exposed to a combination of salt overload and taurine deficiency, no change in sympathetic modulation of vascular tone or in parasympathetic modulation of sinus node function was detected. As for baroreflex sensitivity, in the current study, a tendency to less baroreflex sensitivity was demonstrated in uninephrectomized mice when compared to controls. Likewise, in a model of

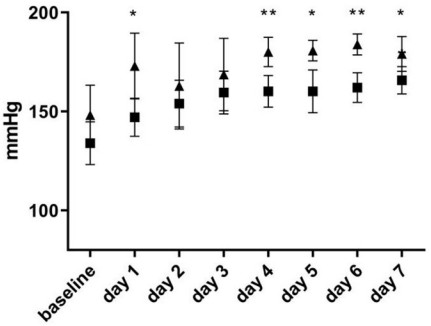

### Peak systolic blood pressure in hypertension model of salt overload and taurin definciency versus angiotensin II

■ uninephrectomy, 2%NaCl, 2%ALA (drinking water)

▲ uninephrectomy, angiotensin II (SC)

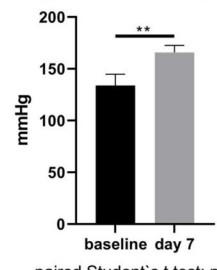

Daily peak systolic arterial pressure in uninephrectomized mice on high-salt diet, taurin deficiency (drinking water: NaCl 2%, L-Ala 2%)

paired Student`s t test: p=0.010

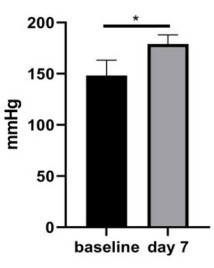

Daily peak systolic arterial pressure in uninephrectomized mice on s.c. Angiotensin-II stimulation

paired Student`s t test: p=0.035

**Fig 4. Upper panel: evolution of daily peak systolic blood pressure in uninephrectomized mice during hypertension-inducing interventions.** a) oral sodium-chloride load and taurine deficiency (NaCl (2%) + L- alanine (2%)-enriched drinking water); b) angiotensin II: 1.4 mg/kg/d subcutaneously applied by osmotic minipump. Lower panel: direct comparison of daily mean systolic blood pressure within groups (baseline versus end of study).

renovascular hypertension with reduced kidney mass, a reduced baroreflex sensitivity was shown previously [19].

As a negative finding, uninephrectomy alone was not associated with arterial hypertension in this study, even though the nominal reduction of resting minimal and maximal heart rate is consistent with an activation of baroreflex to buffer a small rise of blood pressure during the relatively short observation time of 1 week. In a long-term study on uninephrectomized rats, blood pressure was found to be elevated 6 months following uninephrectomy [20]. Conversely, in a model of diabetic-nephropathy, uninephrectomy enhanced the formation of pathophysiologic changes [21]. As for salt overload, the adaptative processes following uninephrectomy include proximal-tubular hypertrophy [22]. A beneficial role of sodium-glucose-cotransporter-2 (SGLT-2) inhibition was shown in uninephrectomized KK/Ay type 2 diabetic mice providing indirect evidence for upregulated SGLT-2, when kidney mass is reduced [23]. In addition, kidney-mass reduction was shown to translate into a higher oxidative stress [24, 25]. The detailed mechanisms, how kidney-mass reduction may increase oxidative stress or reduce anti-oxidant stress defense, still need to be elucidated. As therapeutic measures, both exercise training [26] and use of SGLT-2 inhibitors [27] were shown to reduce oxidative stress in animal models. The Dapagliflozin and Prevention of Adverse Outcomes in Chronic Kidney Disease Study [28], a dedicated study on the use of SGLT-2 inhibitors in chronic kidney disease, clearly showed a benefit for the drug in chronic kidney disease. It remains unclear though, whether or not those interventions are applicable to alleviate oxidative stress following kidney-mass reduction.

As a limitation, a sodium-enriched, taurine-deficient diet was not applied to normal mice without prior uninephrectomy. However, salt overload alone did not increase blood pressure

in an animal model. Rather, a salt overload was shown to predispose to angiotensin-II mediated hypertension [29].

In summary, in the context of uninephrectomy with increased oral salt load and taurine deficiency, arterial hypertension was present. In uninephrectomized animals without increased salt load and taurine deficiency, a reduced vagal modulation of heart rate was found. Future studies conducted over longer observation periods need to determine, whether autonomic changes following uninephrectomy represent a pathomechanism for hypertension. In addition, the present study provides a rationale for experiments on renal salt handling in uninephrectomized mice with or without renal denervation in order to reduce afferent renal autonomic nerve traffic, which, hypothetically, is involved in salt-sensitive hypertension.

## Supporting information

**S1 Data. Control nephrectomy source data.**
(XLS)

## Acknowledgments

The authors thank Günther Amrhein and Susanne Schütze, Vascular Research Centre, Goethe University, Frankfurt (Main), Germany, for technical assistance.

## Author Contributions

**Conceptualization:** Rainer U. Pliquett, Ralf P. Brandes.

**Data curation:** Rainer U. Pliquett.

**Formal analysis:** Rainer U. Pliquett.

**Investigation:** Rainer U. Pliquett.

**Methodology:** Ralf P. Brandes.

**Project administration:** Ralf P. Brandes.

**Resources:** Ralf P. Brandes.

**Software:** Ralf P. Brandes.

**Supervision:** Ralf P. Brandes.

**Validation:** Ralf P. Brandes.

**Writing – original draft:** Rainer U. Pliquett.

**Writing – review & editing:** Rainer U. Pliquett, Ralf P. Brandes.

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
