## [Decision Letter · Decision Letter 0]

15 Nov 2021

PONE-D-21-32299

Experimental uninephrectomy associates with less parasympathetic tone and facilitates sodium-dependent arterial hypertension

PLOS ONE

Dear Dr. Pliquett,

Thank you for submitting your manuscript to PLOS ONE. After careful consideration, we feel that it has merit but does not fully meet PLOS ONE’s publication criteria as it currently stands. Therefore, we invite you to submit a revised version of the manuscript that addresses the points raised during the review process.

Three experienced scientists in this field have reviewed your manuscript. While all three fortunately find it of interest, they also all identify a number of issues that must ALL be dealt with in your revision.

The study design is confusing and should be illustrated with a diagram. The relation between uninephrectomy and blood pressure in your own data should be clarified. The objective and relevance of the comparison between Ang II infusion and high salt/low taurine needs a better justification. This should be based on a clear hypothesis. The role of innervation (of both the excised and the contralateral) kidneys should at least be adequately discussed. The other comments must also be addressed.

We look forward to receiving your revised manuscript.

Kind regards,

Jaap A. Joles, DVM, PhD

Academic Editor

PLOS ONE

Journal Requirements:

"the authors are appreciate the grant support (RUP) by Deutsche Nierenstiftung."

"RUP received grant from "Deutsche Nierenstiftung" 2008 (https://www.nierenstiftung.de). The funders had no role in study design, data collection and analysis, decision to publish, or preparation of the manuscript."

Reviewers' comments:

Reviewer's Responses to Questions

Comments to the Author

1. Is the manuscript technically sound, and do the data support the conclusions?

Reviewer #1: Partly

Reviewer #2: Partly

Reviewer #3: Partly

2. Has the statistical analysis been performed appropriately and rigorously?

Reviewer #1: Yes

Reviewer #2: I Don't Know

Reviewer #3: Yes

3. Have the authors made all data underlying the findings in their manuscript fully available?

Reviewer #1: No

Reviewer #2: No

Reviewer #3: No

4. Is the manuscript presented in an intelligible fashion and written in standard English?

Reviewer #1: No

Reviewer #2: No

Reviewer #3: No

5. Review Comments to the Author

Reviewer #1: Pliquett et al. reported that the parasympathetic tone of uninephrectomized mice was decreased, when compared to that of normal mice without uninephrectomy. In addition, the authors indicated that the levels of blood pressure in uninephrectomized mice with salt overload and taurine deficiency were comparable to those in uninephrectomized mice with angiotensin II infusion. This manuscript is important, because the physiological underpinnings of the blood pressure increase after uninephrectomy remain unclear. However, some serious concerns have been raised.

(1) It is understandable that, as the authors indicated, it takes a long time to increase blood pressure if structural changes in blood vessels are the main cause. However, even though the rapid change of the decrease of parasympathetic nerve was definitely caused, the levels of blood pressure did not change at all. It is difficult for me to understand this discrepancy of the results. The authors should indicate why the levels of blood pressure by uninephrectomy did not change at all more adequately.

(2) There is no data to evaluate the difference between uninephrectomized mice with angiotensin II administration or salt load and those without treatment intervention. The results are important. Therefore, the authors should show the results.

(3) The authors showed the data of HF-HR and LF-SBP in page 7, 3rd paragraph. I do not know when those data were measured. The authors should indicate the time more clearly.

(4) The authors indicated that the evolution of arterial hypertension was similar between sodium chloride overload and angiotensin II stimulation in Figure 3. However, the average daily systolic blood pressure was different on 3 of 7 days under investigation. I think that the levels of systolic blood pressure between sodium chloride overload and angiotensin II stimulation are not same. Therefore, the authors should revise the expression adequately.

(5) Angiotensin II is correct and angiotensin 2 is not correct. Therefore, the authors should revise the word adequately.

(6) The authors indicated that “As a negative finding, uninephrectomy alone did associate with arterial hypertension in this study.” in page 8, 2nd paragraph. I think that uninephrectomy alone did not associate with arterial hypertension in this study. The authors should revise the sentence adequately.

(7) There are some typographical and grammatical errors in this manuscript. Therefore, the manuscript should be reviewed by the native speaker of English.

Reviewer #2: Pliquett and Brandes present a study examining the effect of uninephrectomy on autonomic tone and the development of hypertension. Spectral analysis of blood pressure and heart rate variability was used as a surrogate for autonomic activity, with the authors reporting that uni-nephrectomy had no differences in resting blood pressure, heart rate or spontaneous baroreflex gain. The LF-HF ratio of SBP was shifted higher, and the HF of PI was lower suggesting a shift towards parasympathetic dominance. Two forms of secondary hypertension were induced – via angiotensin II infusion and a high salt/taurine deficient diet. The time course and magnitude of hypertension development was similar between Ang II and high salt/taurine deficiency.

Major Points

The overall object of comparing Ang II and high salt/low taurine hypertension is not clear. The Introduction pitches the heightened cardiovascular risk faced by kidney donors, or those born with one kidney. Is the intension to evaluate the impact of uninephrectomy on the propensity to hypertension when faced with the “challenge” of the two hypertensive models? If so, the main statistical comparisons should be between sham nephrectomy and uninephrectomy, with (1) Ang II and (2) high salt/low taurine. It isn’t clear how comparing the rise in BP between these two quite different experimental models of hypertension gives any insight into the impact of uninephrectomy itself; surely the magnitude of hypertension with Ang II is simply dependant on the dosage. Also, given that the clinical prevalence of low taurine is extremely rare, the real-world relevance of this model needs a clear justification in the text.

The manuscript contains many grammatical, tense and wording errors. Given that PLOS-One does not copy edit, the authors should consider engaging a proof-reader to improve the readability.

Minor Points

The abstract states that blood pressure measurements were made “non-invasively” – the surgical implantation of the telemetry catheter is an invasive procedure. I think what the authors mean is that telemetry recordings were made from conscious, undisturbed animals in their home cages. The term “non-invasive blood pressure” in rodents usually refers to plethysmography.

The opening paragraph of the introduction raises some valid concerns around uninephrectomy but is rather vague – it would be helpful to give some numbers around the increased risk for hypertension in the patient groups mentioned.

The numbers in each group should be clearly stated near the beginning of the Methods.

Given that telemetric recording of BP is a very well-established and commonly used technique, the authors might consider reducing the detail around implantation and give a reference instead?

The authors refer to Mean HR, BP etc being determined over 24h, but elsewhere indicate that recordings were made for 60 minutes each day.

It is more common to express HRV in terms of the LF:HF ratio of PI, rather than HR alone. This is because it is possible for both LF and HF power to change. It is less common to report the LF:HF ratio for SBP as representing “autonomic balance”.

The authors mention “baroreflex curves” – this is not correct for spontaneous baroreflex gain technique they have used; the full curve can generally only be examined using the Oxford technique, whereas sBRG examines ramps in BP (ie generally only on the linear portion of the baroreflex curve).

In paragraph 2 of the discussion, the authors state that “uninephrectomy alone did associate with arterial hypertension” – assume this should be “did not”?

Figures 1 and 2 could be combined, and Figure 3 consider maybe showing just MAP rather than SBP/DBP in separate graphs, and perhaps add in HR?

The data availability statement does not appear compatible with PLOS-One guidelines, as no reason for restricting access to “request from authors” is given.

Reviewer #3: The purpose of this study was to investigate the effect of unilaterial nephrectomy on autonomic tone and arterial blood pressure in mice. Normotensive C57Bl6N mice were subjected to sham surgery or unilateral nephrectomy. Mice were also instrumented with telemetric blood pressure sensors to monitor arterial blood pressure and heart rate. Apparently the nephrectomized mice were treated with subcutaneous infusions of angiotensin II or a sodium chloride-enriched, taurine-deficient diet. The exact protocol is not clear from the Methods section and therefore, the results are difficult to understand and to interpret. Based on spectral analysis of heart rate and blood pressure, the authors conclude that unilateral nephrectomy is associated with reduced parasympathetic tone and that the combination of unilateral nephrectomy and taurine-deficiency predisposes to sodium-chloride sensitive hypertension.

General Comments

The study addresses a potentially important topic. While it is known that patients with only one kidney (e.g., living donor) have an elevated risk for developing hypertension, the exact mechanism for this elevated risk is unknown. The experimental approach with telemetric blood pressure recordings and drug application using osmotic minipumps is technically challenging.

Major Comments

1. The manuscript lacks a specific description of the experimental protocol and results. For example, the exact time course of the protocol is not outlined. It is unclear at which time point after the nephrectomy the data were collected. I think a Figure showing the timeline of the experimental protocol would be helpful. Apparently, the telemetric blood pressure sensors were implanted first and one week later mice were subjected to a second surgery, where the nephrectomy or sham procedure was performed. It is unclear, at which time the osmotic minipumps were implanted and if pumps were replaced after 14 days (it is stated that the mice were observed for 4-6 weeks).

2. Apparently, there were 3 groups (but not 100% sure from the Methods section): (1) A control group without nephrectomy but sham surgery that was not treated with angiotensin or high-salt diet; (2) a unilaterally nephrectomized group with angiotensin II infusion (for how long?); and (3) a unilaterally nephrectomized group with high-salt-low taurine diet. However, Figs. 1 and 2 only show two groups. Is the nephrectomized group the one with angiotension II or with high-salt-low taurine diet? Also Fig. 1 shows no effect on blood pressure, but Fig. 3 shows an increase in blood pressure in the nephrectomized groups. I believe that I don’t understand the protocol and at which time the different data were obtained. I strongly feel that the experimental protocol needs to be described in much more detail.

3. The experimental procedures are technically challenging in mice and mice may not recover easily from these type of interventions (telemetric sensor, osmotic minipumps are relatively large in respect to the small size of the animals). I wonder if more reliable data could have been obtained if the study would have been conducted in larger animals, such as rats. I strongly feel the authors need to explain why mice were used for this study.

4. The Introduction needs a clearly stated hypothesis and rationale why the study was conducted.

5. Page 5, towards the bottom: LF-SBP is not a measure of sympathetic tone. It is a measure of sympathetic modulation of vascular tone. For example, if sympathetic nerve activity is high but there is reduced vascular sympathetic responsiveness, there would be low LF-SBP but high sympathetic tone. Likewise, HF-HR is not a measure of parasympathetic tone. It is a measure of parasympathetic modulation of sinus node function. Again, parasympathetic tone may be high but if the sinus node has reduced parasympathetic responsiveness, there would be reduced HF-HR despite high parasympathetic tone. I suggest changing the wording throughout the manuscript accordingly.

6. Typically, sympathovagal balance is calculated as the ratio of LF-HR to HF-HR. The low frequency component of heart rate variability (not systolic blood pressure variability) is used. The ratio of LF-SBP to HF-HR is comparing sympathetic effects on vascular tone with parasympathetic effects on the sinus node. I don’t think this is a valid marker of sympathovagal balance. Sympathovagal balance should consider the same target organ (i.e., the sinus node).

7. It is unclear where the data from Figs. 1 and 2 come from. One column is labeled as “Uninephrectomized”. However, my understanding of the protocol is that there were two “uninephrectomized” groups. Which one is shown in Figs. 1 and 2 and why is the other uninephrectomized group not shown? You may include the data from the sham operated group in Fig. 3. All figure legends should list the number of animals for each data point shown.

8. What is causing the decrease in parasympathetic modulation of sinus node function in the mice with unilateral nephrectomy? If the authors propose a role for afferent renal nerve traffic, renal denervation experiments should be performed to test this hypothesis.

Minor Comments

1. Abstract, Results section, 1st line: …were not different “n” uninephrectomized mice … “n” should be “in”.

2. Keywords: Avoid keywords that are already in the title.

3. Page 7, line 6 from bottom: “hypetension” should read “hypertension”.

4. Page 7, bottom, last line: the numbers need units.

5. Page 9, 1st line: … a dedicated study … (not studies).

6. Page 8, line 10 from bottom, “model” not “models”.

7. References: Some of the references are dated. Is there newer literature on some of these topics?

6. PLOS authors have the option to publish the peer review history of their article (what does this mean?). If published, this will include your full peer review and any attached files.

Do you want your identity to be public for this peer review? For information about this choice, including consent withdrawal, please see our Privacy Policy.

Reviewer #1: No

Reviewer #2: No

Reviewer #3: No

---

## [Author Response · Author response to Decision Letter 0]

15 Dec 2021

Response to Reviewers, manuscript PONE-D-21-32299

title: „Experimental uninephrectomy associates with less parasympathetic tone and facilitates sodium-dependent arterial hypertension“

PLOS ONE

Reviewers' comments:

Reviewer #1: 

Pliquett et al. reported that the parasympathetic tone of uninephrectomized mice was decreased, when compared to that of normal mice without uninephrectomy. In addition, the authors indicated that the levels of blood pressure in uninephrectomized mice with salt overload and taurine deficiency were comparable to those in uninephrectomized mice with angiotensin II infusion. This manuscript is important, because the physiological underpinnings of the blood pressure increase after uninephrectomy remain unclear. However, some serious concerns have been raised.

(1) It is understandable that, as the authors indicated, it takes a long time to increase blood pressure if structural changes in blood vessels are the main cause. However, even though the rapid change of the decrease of parasympathetic nerve was definitely caused, the levels of blood pressure did not change at all. It is difficult for me to understand this discrepancy of the results. The authors should indicate why the levels of blood pressure by uninephrectomy did not change at all more adequately.

Answer: In Results section, more results on blood-pressure regulation in uninephrectomized mice were added. In Discussion section, these results were discussed accordingly.

New (Page 8, 1st paragraph, last sentence): „However, peak systolic arterial blood pressure tended to be higher in uninephrectomized mice (125.3 ± 20.3 mmHg) as compared to normal controls (114.7 ± 9.5 mmHg), although this difference did not reach the significance level.“

New (Page 8, 2nd paragraph, last sentence):

„However, in uninephrectomized mice, a non-significant tendency to a reduced baroreflex sensitivity of heart rate was demonstrated (5.7 ± 3.6 bpm/mmHg versus 6.8 ± 1.6 bpm/mmHg).“

Old (Discussion, Page 8, 1st paragraph, 1st sentence): „Here, in a murine model of hypertension using a combination of kidney-mass reduction, salt overload and taurine deficiency, the degree of arterial hypertension was comparable to a high-angiotensin model of hypertension.“

New (Discussion, Page 10, 1st paragraph, 1st + 2nd sentence): „Here, in a murine model of kidney-mass reduction, peak systolic arterial blood pressure tended to be higher in comparison to normal controls. In addition, when salt overload and taurine deficiency were applied to uninephrectomized mice, the degree of arterial hypertension was comparable to a high-angiotensin II model of hypertension.“

Old (Discussion, Page 8, 1st paragraph, last sentence): „Changes in baroreflex sensitivity were ruled out between uninephrecomized mice and normal controls here, while a model of renovascular hypertension with reduced kidney mass was associated with a reduced baroreflex sensitivity as shown previously in our group (20).“

New (Discussion, Page 10, 1st paragraph, last 5 lines): „As for baroreflex sensitivity, in the current study, a tendency to less baroreflex sensitivity was demonstrated in uninephrectomized mice when compared to normal controls. Likewise, in a model of renovascular hypertension with reduced kidney mass, a reduced baroreflex sensitivity was shown previously (19).“

(2) There is no data to evaluate the difference between uninephrectomized mice with angiotensin II administration or salt load and those without treatment intervention. The results are important. Therefore, the authors should show the results.

Answer: After sham surgery or after uninephrectomy, we performed 24-hour telemetric recordings 1 week after surgery. Afterwards, in uninephrectomized on salt overload+taurine deficiency or on angiotensin-II treatment, a telemetric 24-hour measurement over 7 consecutive days was performed. Therefore, a direct control group (uninephrectomized mice without intervention) is lacking in this study. However, a comparison of mean and peak systolic arterial pressure at baseline and at the end of study was provided. Methods, Results sections and Figures were revised accordingly.

New: (Methods, Page 6, 1st paragraph, last 3 lines): „For analysis, a within-group comparison of daily mean and peak systolic arterial pressure at baseline and at day 7 was performed“

New (Methods, Page 7, last paragraph, 3rd-4th line): 

„For comparisons within groups, paired Student`s t test was used.“

Old (Results, Page 8, 3rd paragraph, first 2 sentences):

„When uninephrectomized mice were exposed to drinking water with added sodium chloride (2%) and L-alanine (2%), the evolution of arterial hypertension was similar to the evolution of arterial hypetension in a group of uninephrecomized mice subjected to a subcutaneous angiotensin-2 infusion.“

New (Results, Page 8, 3rd paragraph, first 2 sentences): 

„When compared to baseline, uninephrectomized mice subjected to a high-salt and a low-taurine intake had a higher mean systolic arterial pressure (Figure 4) and a higher peak arterial pressure (Figure 5) after one week. Likewise, the positive-control group subjected to angiotensin-II treatment had a higher mean systolic and peak systolic after one week.“

Old: Figure 3

New: Figure 4 + Figure 5

(3) The authors showed the data of HF-HR and LF-SBP in page 7, 3rd paragraph. I do not know when those data were measured. The authors should indicate the time more clearly.

Answer: The times of measurement were specified precisely:

Old (Methods, Page 5, 2nd paragraph, 1st + 2nd line): „After a two-week recovery period, a one-hour baseline recording was taken in the morning in unrestrained, resting conditions. The last 30 minutes were used for analysis.“

New (Methods, Page 6, 2nd paragraph, first 10 lines): „After one week of recovery from sham surgery or uninephrectomy and during the one-week period of secondary hypertension induction, telemetric blood-pressure readings of individual animals were obtained every 5 minutes until the animals were sacrificed. Pulse intervals (defined as consecutive dP/dt) were extracted from aortic blood pressure waveforms using ART 4.2 Gold software (Data Sciences International; St. Paul, Minnesota, USA). Mean heart rate, mean and peak systolic and diastolic blood pressure were determined over 24 hours. In addition, after recovery following sham surgery or uninephrectomy and 1 week after onset of secondary hypertension, a one-hour continuous baseline recording was taken in the morning in unrestrained, resting conditions. The last 30 minutes were used for powerspectral analysis and for determination of spontaneous baroreflex sensitivity as reported previously (13).“

(4) The authors indicated that the evolution of arterial hypertension was similar between sodium chloride overload and angiotensin II stimulation in Figure 3. However, the average daily systolic blood pressure was different on 3 of 7 days under investigation. I think that the levels of systolic blood pressure between sodium chloride overload and angiotensin II stimulation are not same. Therefore, the authors should revise the expression adequately.

Answer: True. The presentation of blood-pressure evolution was revised accordingly.

Old (Results, Page 7, 3rd paragraph, first 3 sentences): „When uninephrectomized mice were exposed to drinking water with added sodium chloride (2%) and L-alanine (2%), the evolution of arterial hypertension was similar to the evolution of arterial hypetension in a group of uninephrecomized mice subjected to a subcutaneous angiotensin-2 infusion. Specifically, average daily systolic pressure was not different on 4 of 7 days under investigation. However, the daily maximum systolic blood pressure yielded from day 4 to day 7 was higher in angiotensin-2 - treated animals (Figure 3).“

New (Results, Page 8, last paragraph, last 3 lines, Page 9, first paragraph, first 2 lines): „Specifically, during the week of hypertension development, daily mean systolic arterial blood pressure was similar in uninephrecomized mice of both groups on 4 of 7 days under investigation. However, on 5 of 7 days, daily peak systolic blood pressure was higher in the positive control group on angiotensin-II - treatment than in the group on salt-rich, taurin-deficient diet.“

(5) Angiotensin II is correct and angiotensin 2 is not correct. Therefore, the authors should revise the word adequately.

Answer: We agree. We corrected this mistake in the manuscript.

(6) The authors indicated that “As a negative finding, uninephrectomy alone did associate with arterial hypertension in this study.” in page 8, 2nd paragraph. I think that uninephrectomy alone did not associate with arterial hypertension in this study. The authors should revise the sentence adequately.

Answer: We agree.

Old (Discussion, Page 8, 2nd paragraph, 1st sentence): „As a negative finding, uninephrectomy alone did associate with arterial hypertension in this study.“

New (Discussion, Page 10, 2nd paragraph, 1st sentence): „As a negative finding, uninephrectomy alone was not associated with arterial hypertension in this study.“

(7) There are some typographical and grammatical errors in this manuscript. Therefore, the manuscript should be reviewed by the native speaker of English.

Answer: The revised version of the manuscript was proof-read by a native speaker (Dr. A. Adeabgo).

Reviewer #2: Pliquett and Brandes present a study examining the effect of uninephrectomy on autonomic tone and the development of hypertension. Spectral analysis of blood pressure and heart rate variability was used as a surrogate for autonomic activity, with the authors reporting that uni-nephrectomy had no differences in resting blood pressure, heart rate or spontaneous baroreflex gain. The LF-HF ratio of SBP was shifted higher, and the HF of PI was lower suggesting a shift towards parasympathetic dominance. Two forms of secondary hypertension were induced – via angiotensin II infusion and a high salt/taurine deficient diet. The time course and magnitude of hypertension development was similar between Ang II and high salt/taurine deficiency.

Major Points

The overall object of comparing Ang II and high salt/low taurine hypertension is not clear.

Answer: As objective, we introduced Ang-II-treated mice as a positive control group for mice on a high-salt and low-taurine intake. We revised Methods and Results section accordingly.

Old (Methods, Page 4, last paragraph, 2nd sentence): „Two groups of mice of 6 animals each were subjected to uninephrectomy and to angiotensin-2-dependent hypertension or to salt overload and taurin deficiency.“

New (Methods, Page 5, last paragraph, lines 4-6): „Afterwards, as an add-on study, uninephrectomized mice were randomly assigned to a model of hypertension due to salt overload and taurine deficiency (n=6) or to angiotensin-II-dependent hypertension model as a positive control group (n=6).“

Old (Results, Page 8, 3rd paragraph, subtitle and first 2 sentences):

„Uninephrectomized mice show similar propensity of hypertension following sodium-chloride overload and following angiotensin-2 stimulation.

When uninephrectomized mice were exposed to drinking water with added sodium chloride (2%) and L-alanine (2%), the evolution of arterial hypertension was similar to the evolution of arterial hypetension in a group of uninephrecomized mice subjected to a subcutaneous angiotensin-2 infusion.“

New (Results, Page 8, 3rd paragraph, subtitle and first 2 sentences): 

„Uninephrectomized mice with sodium-chloride overload and taurine deficiency show a predisposition for hypertension

When compared to baseline, uninephrectomized mice subjected to a high-salt and a low-taurine intake had a higher mean systolic arterial pressure (Figure 4) and a higher peak arterial pressure (Figure 5) after one week. Likewise, the positive-control group subjected to angiotensin-II treatment had a higher mean systolic and peak systolic after one week.“

The Introduction pitches the heightened cardiovascular risk faced by kidney donors, or those born with one kidney. Is the intension to evaluate the impact of uninephrectomy on the propensity to hypertension when faced with the “challenge” of the two hypertensive models? If so, the main statistical comparisons should be between sham nephrectomy and uninephrectomy, with (1) Ang II and (2) high salt/low taurine. 

Answer: It is the intention to point at the sequelae of uninephrectomy in terms of a predisposition for hypertension. As a specific challenge, a high-salt-low-taurine intake was introduced. Ang-II treatment served as a positive control group. Both Introduction and Methods were modified, to convey this study goal.

Old (Introduction, Page 3, 2nd paragraph, 5th sentence): „In the present study, the influence of both uninephrectomy, sodium-chloride overload and taurin deficiency on autonomic nervous system function and on blood-pressure regulation was investigated.“

New (Introduction, Page 3, 3rd paragraph, lines 1-4): „In the present study, the influence of uninephrectomy both on blood-pressure and autonomic-nervous-system regulation was investigated. In addition to uninephrectomy, the role of a hypertensive challenge on autonomic nervous system function and on blood-pressure regulation was investigated. As a hypertensive challenge, a high-salt-low-taurine diet was applied.“

Old (Methods, Page 4, last paragraph, 2nd sentence): „Two groups of mice of 6 animals each were subjected to uninephrectomy and to angiotensin-2-dependent hypertension or to salt overload and taurin deficiency.“

New (Methods, Page 5, 2nd paragraph, lines 4-6): „Afterwards, as an add-on study, uninephrectomized mice were randomly assigned to a model of hypertension due to salt overload and taurine deficiency (n=6) or to angiotensin-II-dependent hypertension model as a positive control group (n=6).“

It isn’t clear how comparing the rise in BP between these two quite different experimental models of hypertension gives any insight into the impact of uninephrectomy itself; surely the magnitude of hypertension with Ang II is simply dependant on the dosage. Also, given that the clinical prevalence of low taurine is extremely rare, the real-world relevance of this model needs a clear justification in the text.

Answer: The impact of uninephrectomy was assessed by hemodynamic data in comparison to sham surgery and by assessing the blood-pressure response to high-salt, low-taurine intake. In the revised manuscript, a comparison of daily mean and peak systolic arterial pressure at baseline versus at end of intervention was performed (revised Figure 4 and Figure 5). The positive control group applying Ang-II treatment in uninephrectomized mice illustrates the magnitude of the hypertensive effect of the intervention (high-salt, low-taurine intake). As for the intervention, a low taurine intake by virtue of high L-alanine intake allows for less taurine-associated attenuation of hypertension. The antihypertensive effect of taurine was demonstrated by Sun and colleagues (reference 12).

The manuscript contains many grammatical, tense and wording errors. Given that PLOS-One does not copy edit, the authors should consider engaging a proof-reader to improve the readability.

The revised version of the manuscript was read by a native speaker (Dr. A. Adeabgo).

Minor Points

The abstract states that blood pressure measurements were made “non-invasively” – the surgical implantation of the telemetry catheter is an invasive procedure. I think what the authors mean is that telemetry recordings were made from conscious, undisturbed animals in their home cages. The term “non-invasive blood pressure” in rodents usually refers to plethysmography.

True. „non-invasively“ is not adequate. Now, this part was revised.

Old (Abstract, Page 2, 2nd paragraph, lines 4-5): „Non-invasive telemetry readings of heart-rate and blood-pressure were obtained.“

New (Abstract, Page 2, 2nd paragraph, lines 4-5): „A transfemoral aortic catheter with telemetry unit was implanted, telemetry readings of heart-rate and blood-pressure were recorded.“

The opening paragraph of the introduction raises some valid concerns around uninephrectomy but is rather vague – it would be helpful to give some numbers around the increased risk for hypertension in the patient groups mentioned.

Answer: We added specific information.

New (Introduction, Page 3, 1st paragraph, lines 4-6): „Specifically, hypertension developed in 4%, 10%, and 51% of kidney donors at 5, 10, and 40 years, respectively (2), and the development of hypertension among kidney donors prevailed after correction for aging (3).”

The numbers in each group should be clearly stated near the beginning of the Methods.

Answer: Patient numbers (per group) were added.

Given that telemetric recording of BP is a very well-established and commonly used technique, the authors might consider reducing the detail around implantation and give a reference instead?

Answer: True. In the revised manuscript, a reference is provided 

Old (Methods, Page 4, 1st paragraph, last 3 sentences): „Telemetry catheters were implanted in all animals as described previously (13). In short, the left femoral artery was chosen for surgical implantation of the aortic catheter. The attached telemetry unit TA11PAC10 (Data Sciences International, St. Paul, Minnesota, USA) was placed subcutaneously on the right flank.“

New (Methods, Page 5, 2nd paragraph, last 3 lines): „Transfemoral aortic catheters with telemetry units (TA11PAC10, Data Sciences International, St. Paul, Minnesota, USA) were implanted in all animals as described previously (13).“

The authors refer to Mean HR, BP etc being determined over 24h, but elsewhere indicate that recordings were made for 60 minutes each day.

Answer: In the revised manuscript, the respective sections were revised:

Old (Methods, Page 5, 2nd paragraph, lines 1-2): „After a two-week recovery period, a one-hour baseline recording was taken in the morning in unrestrained, resting conditions. The last 30 minutes were used for analysis.“

New (Methods, Page 6, 2nd paragraph, lines 1-10): „After one week of recovery from sham surgery or uninephrectomy and during the one-week period of secondary hypertension induction, telemetric blood-pressure readings of individual animals were obtained every 5 minutes until the animals were sacrificed. Pulse intervals (defined as consecutive dP/dt) were extracted from aortic blood pressure waveforms using ART 4.2 Gold software (Data Sciences International; St. Paul, Minnesota, USA). Mean heart rate, mean and peak systolic and diastolic blood pressure were determined over 24 hours. In addition, after recovery following sham surgery or uninephrectomy and 1 week after onset of secondary hypertension, a one-hour continuous baseline recording was taken in the morning in unrestrained, resting conditions. The last 30 minutes were used for powerspectral analysis and for determination of spontaneous baroreflex sensitivity as reported previously (13).“

It is more common to express HRV in terms of the LF:HF ratio of PI, rather than HR alone. This is because it is possible for both LF and HF power to change. It is less common to report the LF:HF ratio for SBP as representing “autonomic balance”.

Answer: LF-SBP:HF-HR ratio is regarded as a measure of sympathovagal balance in mice, as LF:HF ratio derived from powerspectrum of heart rate represents sympatho-vagal balance in humans. In mice, the corresponding frequency bands for sympathetic and vagal tone were published by V. Baudrie and colleagues (Reference 15). We adhered to this methodology. We are aware that other groups determined powerspectral LF- and HF bands in mice, e.g. Chiara Piantoni and colleagues published in Front Neurosci. 2021;15:617698. However, in this publication, only total intensity and parasympathetic intensity of the HF band were reported. The intensities of the LF band of frequency domain were not reported. As an explanation, the LF band does not convey information on sympathetic tone in mice.

The authors mention “baroreflex curves” – this is not correct for spontaneous baroreflex gain technique they have used; the full curve can generally only be examined using the Oxford technique, whereas sBRG examines ramps in BP (ie generally only on the linear portion of the baroreflex curve).

Answer: True. We corrected this mistake.

Old (Methods, Page 5 (last 4 lines), Page 6 (first 5 lines): „Baroreflex sensitivity of heart rate was determined by the sequence technique of concomitant changes of systolic blood pressure and pulse intervals (digitized, linearly interpolated) utilizing the Hemolab software (17-19). Concomitant changes of systolic blood pressure (of at least 15 mmHg) and pulse intervals of at least 4 consecutive heart beats were correlated. For individual baroreflex curves, a correlation coefficient of at least 0.9 was mandated for analysis. In addition, a time delay of 0 seconds was chosen for analysis of concomitant blood-pressure and pulse-interval changes. The average of at least 10 individual baroreflex slopes (linear portion of systolic blood pressure – pulse-interval relationship) was considered as baroreflex sensitivity of heart rate.

New (Methods, Page 7, 1st paragraph, last 9 lines): „Baroreflex sensitivity of heart rate was determined by spontaneous baroreflex gain technique. Sequences of concomitant spontaneous changes of systolic blood pressure (of at least 15 mmHg) and pulse intervals of at least 4 consecutive heart beats were digitized, linearly interpolated and correlated utilizing the Hemolab software (17;18). For individual ramps in blood pressure and pulse intervals, a correlation coefficient of at least 0.9 was set as threshold for data inclusion for the analysis. A time delay of 0 seconds was chosen for analysis of concomitant blood-pressure and pulse-interval changes. The average of at least 10 individual baroreflex ramps (linear portion of systolic blood pressure – pulse-interval relationship) was considered as baroreflex sensitivity of heart rate.“

In paragraph 2 of the discussion, the authors state that “uninephrectomy alone did associate with arterial hypertension” – assume this should be “did not”?

True, this point was corrected.

Old (Discussion, Page 8, 2nd paragraph, 1st sentence): „As a negative finding, uninephrectomy alone did associate with arterial hypertension in this study.“

New (Discussion, Page 10, 2nd paragraph, 1st sentence): „As a negative finding, uninephrectomy alone was not associated with arterial hypertension in this study.“

Figures 1 and 2 could be combined, and Figure 3 consider maybe showing just MAP rather than SBP/DBP in separate graphs, and perhaps add in HR?

In merged Figure 2, a graph on heart rate was added, a graph on LF-SBP:HF-HR ratio was omitted.

The data availability statement does not appear compatible with PLOS-One guidelines, as no reason for restricting access to “request from authors” is given.

Answer: The data availability statement was revised. An ethical issue was named for restricted data access.

Reviewer #3: The purpose of this study was to investigate the effect of unilaterial nephrectomy on autonomic tone and arterial blood pressure in mice. Normotensive C57Bl6N mice were subjected to sham surgery or unilateral nephrectomy. Mice were also instrumented with telemetric blood pressure sensors to monitor arterial blood pressure and heart rate. Apparently the nephrectomized mice were treated with subcutaneous infusions of angiotensin II or a sodium chloride-enriched, taurine-deficient diet. The exact protocol is not clear from the Methods section and therefore, the results are difficult to understand and to interpret. Based on spectral analysis of heart rate and blood pressure, the authors conclude that unilateral nephrectomy is associated with reduced parasympathetic tone and that the combination of unilateral nephrectomy and taurine-deficiency predisposes to sodium-chloride sensitive hypertension.

General Comments

The study addresses a potentially important topic. While it is known that patients with only one kidney (e.g., living donor) have an elevated risk for developing hypertension, the exact mechanism for this elevated risk is unknown. The experimental approach with telemetric blood pressure recordings and drug application using osmotic minipumps is technically challenging.

Major Comments

1. The manuscript lacks a specific description of the experimental protocol and results. For example, the exact time course of the protocol is not outlined. It is unclear at which time point after the nephrectomy the data were collected. I think a Figure showing the timeline of the experimental protocol would be helpful. Apparently, the telemetric blood pressure sensors were implanted first and one week later mice were subjected to a second surgery, where the nephrectomy or sham procedure was performed. It is unclear, at which time the osmotic minipumps were implanted and if pumps were replaced after 14 days (it is stated that the mice were observed for 4-6 weeks).

Answer: Osmotic minipumps were not changed as stated on Page 4, last paragraph, 6th line („The osmotic minipumps were used to a maximum duration of administration of 14 days.“). The respective statement that mice were observed for 4-6 weeks was revised.

Old (Methods, Page 5, 1st and 2nd line): „After observing the mice for 4 - 6 weeks, the mice were sacrificed under isoflurane anaesthesia by decapitation.“

New (Methods, Page 6, 1st paragraph, lines 5-6): „After finishing the experiments, the mice were sacrificed by decapitation under isoflurane anaesthesia.“

2. Apparently, there were 3 groups (but not 100% sure from the Methods section): (1) A control group without nephrectomy but sham surgery that was not treated with angiotensin or high-salt diet; (2) a unilaterally nephrectomized group with angiotensin II infusion (for how long?); and (3) a unilaterally nephrectomized group with high-salt-low taurine diet. However, Figs. 1 and 2 only show two groups. Is the nephrectomized group the one with angiotension II or with high-salt-low taurine diet? Also Fig. 1 shows no effect on blood pressure, but Fig. 3 shows an increase in blood pressure in the nephrectomized groups. I believe that I don’t understand the protocol and at which time the different data were obtained. I strongly feel that the experimental protocol needs to be described in much more detail.

Answer: Figure 1 showing the sequence of interventions was added. There were 2 groups: (1) normal controls without nephrectomy, but sham surgery, (2) uninephrectomized mice. In an add-on experiment, uninephrectomized mice were randomly assigned to a high-salt-low-taurine diet or to an angiotensin-II treatment. Method section was revised accordingly.

Old (Methods, Page 4, lines 11-19): „Telemetry catheters were implanted in all animals as described previously (13). In short, the left femoral artery was chosen for surgical implantation of the aortic catheter. The attached telemetry unit TA11PAC10 (Data Sciences International, St. Paul, Minnesota, USA) was placed subcutaneously on the right flank.

One week later, one group of mice was subjected to a sham surgery (skin incision plus manipulation on one kidney). Two groups of mice of 6 animals each were subjected to uninephrectomy and to angiotensin-2-dependent hypertension or to salt overload and taurin deficiency.“

New (Methods, Page 5, 1st paragraph, lines 8-9 and lines 12-14, 2nd paragraph, lines 1-6):

„The experimental setting was displayed in Figure 1. ... Transfemoral aortic catheters with telemetry units (TA11PAC10, Data Sciences International, St. Paul, Minnesota, USA) were implanted in all animals as described previously (13).

One week later, one group of mice was subjected to a sham surgery (n=10) which consisted of skin incision plus manipulation of one kidney. Another group was subjected to uninephrectomy (n=15). After one week of recovery, hemodynamic parameters were obtained and powerspectral analysis performed as outlined below. Afterwards, as an add-on study, uninephrectomized mice were randomly assigned to a model of hypertension due to salt overload and taurine deficiency (n=6) or to angiotensin-II-dependent hypertension model as a positive control group (n=6).“

3. The experimental procedures are technically challenging in mice and mice may not recover easily from these type of interventions (telemetric sensor, osmotic minipumps are relatively large in respect to the small size of the animals). I wonder if more reliable data could have been obtained if the study would have been conducted in larger animals, such as rats. I strongly feel the authors need to explain why mice were used for this study.

True. Physiological studies on kidney-mass reduction need to be performed in larger rodents and other animal models. Mice were chosen to be able to perform add-on studies on genetically modified animals with and without kidney-mass reduction.

4. The Introduction needs a clearly stated hypothesis and rationale why the study was conducted.

Answer: We agree.

New (Introduction, Page 3: last 2 lines, Page 4, lines 1-2): „Here we hypothesized that, firstly, blood pressure is higher in uninephrectomized mice than in normal controls. Secondly, uninephrectomy correlates with a predisposition for hypertension after a high-salt-low-taurine intake and also with changes in sympathetic and/or parasympathetic tone, when compared with normal controls.“

5. Page 5, towards the bottom: LF-SBP is not a measure of sympathetic tone. It is a measure of sympathetic modulation of vascular tone. For example, if sympathetic nerve activity is high but there is reduced vascular sympathetic responsiveness, there would be low LF-SBP but high sympathetic tone. Likewise, HF-HR is not a measure of parasympathetic tone. It is a measure of parasympathetic modulation of sinus node function. Again, parasympathetic tone may be high but if the sinus node has reduced parasympathetic responsiveness, there would be reduced HF-HR despite high parasympathetic tone. I suggest changing the wording throughout the manuscript accordingly.

Answer: We agree. Changes were implemented accordingly.

6. Typically, sympathovagal balance is calculated as the ratio of LF-HR to HF-HR. The low frequency component of heart rate variability (not systolic blood pressure variability) is used. The ratio of LF-SBP to HF-HR is comparing sympathetic effects on vascular tone with parasympathetic effects on the sinus node. I don’t think this is a valid marker of sympathovagal balance. Sympathovagal balance should consider the same target organ (i.e., the sinus node).

Answer: Both frequency domains (LF-SBP, HF-HR) of powerspectral analysis correlated with sympathetic modulation and parasympathetic modulation in mice (Baudrie and colleagues (Reference 15)). Therefore, we introduce the ratio LF-SBP:HF-HR as a surrogate of the classic LF:HF ratio, i.e. sympathovagal balance, in humans. LF band of the frequency domain does not convey information on sympathetic modulation in mice.

7. It is unclear where the data from Figs. 1 and 2 come from. One column is labeled as “Uninephrectomized”. However, my understanding of the protocol is that there were two “uninephrectomized” groups. Which one is shown in Figs. 1 and 2 and why is the other uninephrectomized group not shown? You may include the data from the sham operated group in Fig. 3. All figure legends should list the number of animals for each data point shown.

Answer: The new Figure 1 and the revised Methods section clearly describe the 2 groups displayed in the merged Figure 2 (formerly Figure 1 and 2). We gratefully accept this point.

8. What is causing the decrease in parasympathetic modulation of sinus node function in the mice with unilateral nephrectomy? If the authors propose a role for afferent renal nerve traffic, renal denervation experiments should be performed to test this hypothesis.

Answer: Discussion section was expanded accordingly. 

New (Discussion, Page 11, last 4 lines): „The present study provides a rationale for experiments on renal salt handling in uninephrectomized mice with or without renal denervation in order to reduce afferent renal autonomic nerve traffic, which, hypothetically, is involved in salt-sensitive hypertension there.“

Minor Comments

1. Abstract, Results section, 1st line: …were not different “n” uninephrectomized mice … “n” should be “in”. Corrected.

2. Keywords: Avoid keywords that are already in the title. Keywords were revised accordingly.

3. Page 7, line 6 from bottom: “hypetension” should read “hypertension”. Corrected.

4. Page 7, bottom, last line: the numbers need units. Corrected.

5. Page 9, 1st line: … a dedicated study … (not studies). Corrected.

6. Page 8, line 10 from bottom, “model” not “models”. Corrected.

7. References: Some of the references are dated. Is there newer literature on some of these topics?

Answer: One reference was removed (Ref. 18), one reference (Ref. 24) was replaced by a more relevant one (Tahara A et al.). Discussion was revised accordingly:

Old (Page 8, 3rd paragraph, 1st sentence): „As for salt overload, the adaptative processes following uninephrectomy including proximal-tubular hypertrophy (23) may translate into a higher sodium reabsorption (24).“

New (Page 10, 2nd paragraph, lines 5-9): „As for salt overload, the adaptative processes following uninephrectomy include proximal-tubular hypertrophy (22). A beneficial role of sodium-glucose-cotransporter-2 (SGLT-2) inhibition was shown in uninephrectomized KK/Ay type 2 diabetic mice providing indirect evidence that SGLT-2 is upregulated when kidney mass is reduced (23).“

---

## [Decision Letter · Decision Letter 1]

20 Jan 2022

PONE-D-21-32299R1Experimental uninephrectomy associates with less parasympathetic modulation of heart rate and facilitates sodium-dependent arterial hypertensionPLOS ONE

Dear Dr. Pliquett,

Thank you for submitting your manuscript to PLOS ONE. After careful consideration, we feel that it has merit but does not fully meet PLOS ONE’s publication criteria as it currently stands. Therefore, we invite you to submit a revised version of the manuscript that addresses the points raised during the review process.

 The three reviewers have evaluated your revised manuscript. All three find that the revised paper is better than the original manuscript and so do I.However, only  #1 is completely satisfied.Reviewers #2 and reviewer #3 still raise a number of important points that need attention.Considering the number of issues and the recommendation of both reviewers I have chosen major revision and not minor revision. We look forward to a further revised manuscript. Please submit your revised manuscript by Mar 06 2022 11:59PM. If you will need more time than this to complete your revisions, please reply to this message or contact the journal office at plosone@plos.org. Please include the following items when submitting your revised manuscript:A rebuttal letter that responds to each point raised by the academic editor and reviewer(s). You should upload this letter as a separate file labeled 'Response to Reviewers'.A marked-up copy of your manuscript that highlights changes made to the original version. You should upload this as a separate file labeled 'Revised Manuscript with Track Changes'.An unmarked version of your revised paper without tracked changes. You should upload this as a separate file labeled 'Manuscript'.

We look forward to receiving your revised manuscript.

Kind regards,

Jaap A. Joles, DVM, PhD

Academic Editor

PLOS ONE

Reviewers' comments:

Reviewer's Responses to Questions

**Comments to the Author**

1. If the authors have adequately addressed your comments raised in a previous round of review and you feel that this manuscript is now acceptable for publication, you may indicate that here to bypass the “Comments to the Author” section, enter your conflict of interest statement in the “Confidential to Editor” section, and submit your "Accept" recommendation.

Reviewer #1: (No Response)

Reviewer #2: (No Response)

Reviewer #3: (No Response)

2. Is the manuscript technically sound, and do the data support the conclusions?

Reviewer #1: Yes

Reviewer #2: Partly

Reviewer #3: Partly

3. Has the statistical analysis been performed appropriately and rigorously? 

Reviewer #1: Yes

Reviewer #2: Yes

Reviewer #3: Yes

4. Have the authors made all data underlying the findings in their manuscript fully available?

Reviewer #1: Yes

Reviewer #2: No

Reviewer #3: Yes

5. Is the manuscript presented in an intelligible fashion and written in standard English?

Reviewer #1: Yes

Reviewer #2: Yes

Reviewer #3: No

6. Review Comments to the Author

Reviewer #1: Because the authors have made adequate corrections based on my advice as much as possible, I have no special concerns in the revised manuscript.

Reviewer #2: Thank you to the authors for the considerable revisions to this paper, which have greatly improved clarity and readability. In particular, the scientific writing is much improved, which makes reviewing the manuscript markedly easier. There remain some significant issues with the experimental design and interpretation of results.

Major:

The statement “uninephrectomy and taurine-deficiency predisposes to salt-sensitive hypertension” is not supported by the data. To demonstrate that uninephrectomy makes a subject more vulnerable to hypertension, the authors would need to show that taurine-deficiency and high salt do not induce hypertension in a sham group (or that the magnitude of hypertension was less, or the onset delayed in sham animals).

There remains no clear reason to include the Ang II “positive control” group. The authors state that “during the week of hypertension development…mean BP was similar (between Ang II and taurine/NaCl) on 4/7 days, but peak BP was higher in 5/7 days”. This is not a meaningful comparison, as the rate of onset and the degree of hypertension with Ang II is almost entirely dependant on the dose given – repeating this study with a lower or higher dose of Ang II selected would almost certainly give different findings. If the comparison were instead between an identical dose of Ang II in uninephrectomized vs sham groups, then the authors could draw some valid conclusions about the susceptibility to Ang II hypertension.

I believe that the authors have misinterpreted how to apply the literature on the spectral analysis of blood pressure and heart rate in mice and invite them to carefully reconsider (and potentially re-analyze) their results. The papers cited (Baudrie, 2007 with an editorial commentary by Stauss, 2007) are investigations of the appropriate frequency bands specific to mice, including validation using various drugs to block components of the autonomic nervous system. The authors are correct that the literature most clearly supports the LF(SBP) power (0.15-0.6 Hz) generally correlating with sympathetic vasomotor tone, and the HF(PI) power (2.5-5 Hz) with cardiac vagal tone (largely respiratory). However, it is definitely not appropriate to combine the power outputs from SBP and PI into a single LF:HF ratio. I cannot find any examples of this in the literature, in any species. If the authors wish, they may like to report the LF:HF ratios for both SBP and PI separately. But in general, the evidence is more supportive of the LF:HF ratio for PI being the most appropriate indication of cardiac autonomic balance – the LF:HF ratio for SBP is rarely reported (unlike LF(SBP) alone). Thireau et al have published a guide on performing spectral analysis in mice (https://physoc.onlinelibrary.wiley.com/doi/10.1113/expphysiol.2007.040733) or the authors could refer to many more recent publications where frequency domain analyses of HR and BP in mice have been published (eg:

https://www.ncbi.nlm.nih.gov/pmc/articles/PMC5989555/, https://www.nature.com/articles/s41598-017-17690-7, https://www.frontiersin.org/articles/10.3389/fnins.2021.617698/full, https://physoc.onlinelibrary.wiley.com/doi/full/10.14814/phy2.12811).

Minor

Introduction “hypertension developed…in kidney donors at 5, 10 and 40 years” should add “years post-transplant” (assuming this is what is meant?) to avoid confusion with the preceding sentence about children born with 1 kidney developing hypertension by the age of 18.

The methods now read “telemetric blood pressure readings…were obtained every 5 minutes until animals were sacrificed”. Could the authors simply state how long each scheduled recording of BP was for? Ie 30s every 5 mins

Reviewer #3: The revised manuscript has been improved. Specifically, adding the new Fig. 1 helps to better understand the protocol. I still have a couple of issues that should be addressed:

(comments and page numbers refer to the red-lined version)

1. Abstract, Methods section: “A transfemoral aortic catheter with telemetric unit was implanted “and” telemetry readings …”

2. Abstract, Results and Conclusions: These sections still refer to “parasympathetic tone”, which was not really assessed in this study. Please use the term “parasympathetic modulation of sinus node function” instead.

3. Page 4, Methods, subsection “Animal Models”: I think Darmstadt is in Hessen (not Hesse, although the local dialect in Darmstadt may call it “Hesse”).

4. Throughout the manuscript: present vs. past tense does not appear to be used properly.

5. Page 6 top: How can the pulse interval be defined as “consecutive dP/dt”? Maybe dP/dtmax was used to identify the start and end of individual heart beats?

6. Page 6 bottom: “quantitative measure of sympathetic modulation” should read “… of sympathetic modulation of vascular tone”. Likewise “parasympathetic modulation of sinus node function” (not just modulation).

7. Again, I disagree with the use of the LF-SBP to HF-HR ratio. This ratio does not make any sense physiologically. You are building a ration of responses in different target organs. Furthermore, this ratio is not really needed for the purpose of this study. Since there was no indication of any effects on sympathetic modulation of vascular tone, the use of a ratio is not needed. You can simply use the measured of parasympathetic modulation of sinus node function. With this regard, I think you should calculate and report RMSSD to supplement the frequency-domain analysis and confirm the HF spectral power data.

8. Figure 2: The values on the y-axis for LF-SBP are wrong. A spectral power of 80,000 mmHg2 would correspond to an amplitude of blood pressure fluctuations of 280 mmHg !!! (not possible). Please check your numbers. I think you forgot to multiply with the spectral resolution (but I am not sure). Anyway, these data are wrong.

9. Throughout the Results section the references to Figure numbers do not match with the actual Figures.

10. The first paragraph of the Discussion is redundant and simply repeats the Results. This paragraph should be deleted and replaced with a meaningful discussion of the data.

11. Summary paragraph at the end of the Discussion: Capital T for the sentence starting in “The present study …”. Last word of this paragraph “… hypertension there”. I don’t think you need the word “there”.

12. If uninephrectomy really were associated with less cardiac parasympathetic tone, why then was heart rate not higher?

7. PLOS authors have the option to publish the peer review history of their article (what does this mean?). If published, this will include your full peer review and any attached files.

Reviewer #1: No

Reviewer #2: No

Reviewer #3: **Yes: **Harald M. Stauss

---

## [Author Response · Author response to Decision Letter 1]

9 Feb 2022

Reviewer #2: Thank you to the authors for the considerable revisions to this paper, which have greatly improved clarity and readability. In particular, the scientific writing is much improved, which makes reviewing the manuscript markedly easier. There remain some significant issues with the experimental design and interpretation of results.

We thank the reviewer for reevaluating our manuscript. Below, please find our point-by-point reply.

The statement “uninephrectomy and taurine-deficiency predisposes to salt-sensitive hypertension” is not supported by the data. To demonstrate that uninephrectomy makes a subject more vulnerable to hypertension, the authors would need to show that taurine-deficiency and high salt do not induce hypertension in a sham group (or that the magnitude of hypertension was less, or the onset delayed in sham animals).

We agree with this statement. A sham group was not used in this study. The conclusions (Abstract), the Introduction and the Discussion have been changed accordingly:

Old (Page 2, Abstract, Conclusions):

„The combination of uninephrectomy and taurine-deficiency predisposes to sodium-chloride sensitive hypertension.“

New (Page 2, Abstract, Conclusions):

„The combination of uninephrectomy, taurine-deficiency and high-salt intake led to arterial hypertension.“

Old (Introduction, Page 3 (last 2 lines) and 4 (first 2 lines)):

„Here we hypothesized that, firstly, blood pressure is higher in uninephrectomized mice than in normal controls. Secondly, uninephrectomy correlates with a predisposition for hypertension after a high-salt-low-taurine intake and also with changes in sympathetic and/or parasympathetic tone, when compared with normal controls.“

New (Introduction, Page 3 (last 2 lines) and 4 (first 2 lines)):

„Here we hypothesized, firstly, that blood pressure is higher in uninephrectomized mice than in normal controls, and secondly, that uninephrectomy leads to changes in sympathetic tone and/or in parasympathetic modulation of sinus node function when compared with normal controls. In addition, we hypothesized that salt overload with taurine deficiency increases systolic blood pressure in uninephrectomized mice to a comparable extent as in hypertensive mice on a standard dose of subcutaneous angiotensin-2 stimulation.“

Old (Discussion, Page 11, 2nd + 3rd paragraph):

„However, to the best of our knowledge, salt overload alone did not increase blood pressure in an animal model. Rather, a sodium- chloride overload was shown to predispose to angiotensin-II mediated hypertension (29).

In summary, in uninephrectomized mice, the compensatory increase of sodium-chloride reabsorption by the remaining kidney may predispose to arterial hypertension in the context of an increased oral salt load and taurine deficiency. The reduced vagal modulation of heart rate may represent another underlying pathomechanism for hypertension following uninephrectomy. The present study provides a rationale for experiments on renal salt handling in uninephrectomized mice with or without renal denervation in order to reduce afferent renal autonomic nerve traffic, which, hypothetically, is involved in salt-sensitive hypertension there.“

New (Discussion, Page 11, 2nd + 3rd paragraph):

„However, salt overload alone did not increase blood pressure in an animal model. Rather, a salt overload was shown to predispose to angiotensin-II mediated hypertension (29).

In summary, in the context of uninephrectomy with increased oral salt load and taurine deficiency, arterial hypertension was present. In uninephrectomized animals without increased salt load and taurine deficiency, a reduced vagal modulation of heart rate was found. Future studies conducted over longer observation periods need to determine, whether autonomic changes following uninephrectomy represent a pathomechanism for hypertension. In addition, the present study provides a rationale for experiments on renal salt handling in uninephrectomized mice with or without renal denervation in order to reduce afferent renal autonomic nerve traffic, which, hypothetically, is involved in salt-sensitive hypertension.“

There remains no clear reason to include the Ang II “positive control” group. The authors state that “during the week of hypertension development…mean BP was similar (between Ang II and taurine/NaCl) on 4/7 days, but peak BP was higher in 5/7 days”. This is not a meaningful comparison, as the rate of onset and the degree of hypertension with Ang II is almost entirely dependant on the dose given – repeating this study with a lower or higher dose of Ang II selected would almost certainly give different findings. If the comparison were instead between an identical dose of Ang II in uninephrectomized vs sham groups, then the authors could draw some valid conclusions about the susceptibility to Ang II hypertension.

We agree that the design of these study groups does not allow for a comparison as far as susceptibility to Ang II is concerned. Therefore, any judgement on the degree of hypertension reached by high-salt, taurine-deficient diet was avoided. The term „similar“ was removed from the abstract, the Discussion was modified:

Old (Page 9, Discussion, lines 2-6):

„In addition, when salt overload and taurine deficiency were applied to uninephrectomized mice, the degree of arterial hypertension was comparable to a high-angiotensin II model of hypertension. While the high – angiotensin II model was associated with sympathoactivation, the parasympathetic modulation of heart rate was decreased in uninephrectomized mice, as compared to normal mice without a reduced kidney mass.“

New (Page 9, Discussion, lines 2-6):

„While the high – angiotensin II model associated with sympathoactivation, uninephrectomized mice showed less parasympathetic modulation of sinus node function as compared to normal controls without a reduced kidney mass. In addition, when salt overload and taurine deficiency were applied to uninephrectomized mice, arterial hypertension developed within the same time frame as in the angiotensin-II model of hypertension.“

In the revised manuscript, we described the evolution of hypertension in uninephrectomized mice on a high-salt and taurine-deficient diet in comparison to the established Ang II model of hypertension using a standard dose of Ang II. Specifically, we used the dose of Ang II (1.4 mg per kg body weight), as published previously:…I believe that the authors have misinterpreted how to apply the literature on the spectral analysis of blood pressure and heart rate in mice and invite them to carefully reconsider (and potentially re-analyze) their results. The papers cited (Baudrie, 2007 with an editorial commentary by Stauss, 2007) are investigations of the appropriate frequency bands specific to mice, including validation using various drugs to block components of the autonomic nervous system. The authors are correct that the literature most clearly supports the LF(SBP) power (0.15-0.6 Hz) generally correlating with sympathetic vasomotor tone, and the HF(PI) power (2.5-5 Hz) with cardiac vagal tone (largely respiratory). However, it is definitely not appropriate to combine the power outputs from SBP and PI into a single LF:HF ratio. I cannot find any examples of this in the literature, in any species. If the authors wish, they may like to report the LF:HF ratios for both SBP and PI separately. But in general, the evidence is more supportive of the LF:HF ratio for PI being the most appropriate indication of cardiac autonomic balance – the LF:HF ratio for SBP is rarely reported (unlike LF(SBP) alone). Thireau et al have published a guide on performing spectral analysis in mice (https://physoc.onlinelibrary.wiley.com/doi/10.1113/expphysiol.2007.040733) or the authors could refer to many more recent publications where frequency domain analyses of HR and BP in mice have been published

Thank you for this comment on the methodology used to determine surrogates of sympathetic tone and on parasympathetic modulation of heart rate. Now, we omitted the presentation of sympathetic-parasympathetic ratio (LF-SBP/HR-HF). The protocol by Thireau et al. stated (under the Heading „Short-term ECG analysis“ between Figure 4 and Figure 5) that “LF and HF were significantly diminished by atropine (LF, 2.48 ± 0.23 versus 0.89 ± 0.12 ms2; and HF, 2.82 ± 0.34 versus 0.94 ± 0.11 ms2, P < 0.01, before and after injection, respectively).“

Therefore, in mice, the cumulative intensity of the LF-band of the powerspectrum of heart rate was not used as a surrogate of sympathetic modulation of heart rate. In fact, in mice, the LF-band of heart rate is regarded as a means to gauge parasympathetic modulation of heart rate, e.g. by Dr. Volkmar Gross (Max-Delbrück-Centre, Berlin, Germany) for a LF band between 0.25–1.0 Hz): Costa-Goncalves AC, Tank J, Plehm R, Diedrich A, Todiras M, Gollasch M, Heuser A, Wellner M, Bader M, Jordan J, Luft FC, Gross V. Role of the multidomain protein spinophilin in blood pressure and cardiac function regulation. Hypertension. 2008; 52: 702–707. There, the authors state (under the Heading „Autonomic Imbalance in db/db Mice“): „Low-frequency systolic BP (LF-SBP) oscillations are produced by sympathetic modulation of vascular tone. ...In mice, low-frequency HR oscillations are mediated through parasympathetic mechanisms.19 In contrast, in human subjects, low-frequency HR oscillations result from sympathetic and parasympathetic activation.“

Obst M, Tank J, Plehm R, Blumer KJ, Diedrich A, Jordan J, Luft FC, Gross V. NO-dependent blood pressure regulation in RGS2-deficient mice. Am J Physiol Regul Integr Comp Physiol. 2006 Apr;290(4):R1012-9. doi: 10.1152/ajpregu.00288.2005. Epub 2005 Nov 3. There, the authors reflect the topic of parasympathetic control of the LF-band in Figure 5.

Likewise, a different group (Véronique Baudrie et al. Am J Physiol Regul Integr Comp Physiol 292: R904–R912, 2007) reported, that „Atropine also significantly reduced the LF component of the PI spectrum.“

In the current study, we used the HF band (between 2.5 and 5 Hz) as a measure of parasympathetic modulation of heart reate in accordance with the guidance provided by Thireau et al. (https://physoc.onlinelibrary.wiley.com/doi/10.1113/expphysiol.2007.040733)

With respect to the following references: https://www.ncbi.nlm.nih.gov/pmc/articles/PMC5989555/, https://www.nature.com/articles/s41598-017-17690-7, https://www.frontiersin.org/articles/10.3389/fnins.2021.617698/full There, an ECG was obtained allowing for R-wave detection. As we used pulse intervals calculated by intervals between consecutive blood pressure peaks (first derivative of peak systolic blood pressure or dp/dtmax) the following reference applies to the revised manuscript: https://physoc.onlinelibrary.wiley.com/doi/full/10.14814/phy2.12811)

There, the methodology for sympathovagal balance of the heart cited the following reference:

Montano, N., T. G. Ruscone, A. Porta, F. Lombardi, M. Pagani, and A. Malliani. 1994. Power spectrum analysis of heart rate variability to assess the changes in sympathovagal balance during graded orthostatic tilt. Circulation 90:1826–1831. This reference (a study performed in human volunteers) is not applicable to mice for the reasons mentioned above (LF-band of heart rate does not represent sympathetic modulation of heart rate in mice).

Minor

Introduction “hypertension developed…in kidney donors at 5, 10 and 40 years” should add “years post-transplant” (assuming this is what is meant?) to avoid confusion with the preceding sentence about children born with 1 kidney developing hypertension by the age of 18.

Thank you, we changed this sentence accordingly.

The methods now read “telemetric blood pressure readings…were obtained every 5 minutes until animals were sacrificed”. Could the authors simply state how long each scheduled recording of BP was for? Ie 30s every 5 mins

We agree and implemented this change.

Old (Page 6, 2nd paragraph, first sentence):

“After one week of recovery from sham surgery or uninephrectomy and during the one-week period of secondary hypertension induction, telemetric blood-pressure readings of individual animals were obtained every 5 minutes for until the animals were sacrificed.“

New (Page 6, 2nd paragraph, first sentence):

“After one week of recovery from sham surgery or uninephrectomy and during the one-week period of secondary hypertension induction, telemetric blood-pressure readings of individual animals were obtained every 5 minutes for 10 seconds until the animals were sacrificed.”

Reviewer #3: The revised manuscript has been improved. Specifically, adding the new Fig. 1 helps to better understand the protocol. I still have a couple of issues that should be addressed:

(comments and page numbers refer to the red-lined version)

1. Abstract, Methods section: “A transfemoral aortic catheter with telemetric unit was implanted “and” telemetry readings …”

Thank you. The 2nd mention of „telemetry“ was deleted.

2. Abstract, Results and Conclusions: These sections still refer to “parasympathetic tone”, which was not really assessed in this study. Please use the term “parasympathetic modulation of sinus node function” instead.

This correction was made throughout the text of the revised manuscript.

3. Page 4, Methods, subsection “Animal Models”: I think Darmstadt is in Hessen (not Hesse, although the local dialect in Darmstadt may call it “Hesse”).

The name of the federal state was changed accordingly, thanks.

4. Throughout the manuscript: present vs. past tense does not appear to be used properly.

We revised the manuscript accordingly.

5. Page 6 top: How can the pulse interval be defined as “consecutive dP/dt”? Maybe dP/dtmax was used to identify the start and end of individual heart beats?

Correct, the manuscript was changed accordingly.

6. Page 6 bottom: “quantitative measure of sympathetic modulation” should read “… of sympathetic modulation of vascular tone”. Likewise “parasympathetic modulation of sinus node function” (not just modulation).

Thank you for this point, we changed the text accordingly.

7. Again, I disagree with the use of the LF-SBP to HF-HR ratio. This ratio does not make any sense physiologically. You are building a ration of responses in different target organs. Furthermore, this ratio is not really needed for the purpose of this study. Since there was no indication of any effects on sympathetic modulation of vascular tone, the use of a ratio is not needed. You can simply use the measured of parasympathetic modulation of sinus node function. With this regard, I think you should calculate and report RMSSD to supplement the frequency-domain analysis and confirm the HF spectral power data.

Thank you, the term sympathovagal balance and the respective data were omitted. Time-domain analysis including HRV, RMSSD was not performed in this study due to the relatively short continuous recordings (over 1 hour) for data analysis. Except for those continuous data measurements for analysis for frequency domain data, the telemetric blood pressure readings were obtained every 5 minutes for 10 seconds. To perform time-domain analysis in mice, continuous heart-rate data would be needed over 24 hours (Thireau et al, https://physoc.onlinelibrary.wiley.com/doi/10.1113/expphysiol.2007.040733).

8. Figure 2: The values on the y-axis for LF-SBP are wrong. A spectral power of 80,000 mmHg2 would correspond to an amplitude of blood pressure fluctuations of 280 mmHg !!! (not possible). Please check your numbers. I think you forgot to multiply with the spectral resolution (but I am not sure). Anyway, these data are wrong.

The calculation of LF-SBP was performed as outlined in Methods. We used Dataquest ART 4.2 Gold software and adhered to the Methodology provided by the manufacturer (Data Sciences International). Data analysis was performed in the same way as reported previously (Reference 13 in the manuscript). The difference in the range of the data (Y axis) to data reported elsewhere may be due to differences in interpolation of systolic blood-pressure data performed prior to Fourier transformation when using different software packages.

9. Throughout the Results section the references to Figure numbers do not match with the actual Figures.

The corrections were made accordingly, thank you.

10. The first paragraph of the Discussion is redundant and simply repeats the Results. This paragraph should be deleted and replaced with a meaningful discussion of the data.

The first paragraph was re-written, redundancies were removed.

11. Summary paragraph at the end of the Discussion: Capital T for the sentence starting in “The present study …”. Last word of this paragraph “… hypertension there”. I don’t think you need the word “there”.

Change were made as pointed out, thanks.

12. If uninephrectomy really were associated with less cardiac parasympathetic tone, why then was heart rate not higher?

With exception of a tendency to (higher) maximum blood pressure, hemodynamic data including heart rate were not different at the end of this short-term study.

Now, nominal heart rate has been reported in the Results section to point at a slight decrease of the minimal and maximal heart rate in uninephrectomized mice under resting conditions likely due to baroreflex activation to buffer a small increase of blood pressure.

New (Page 8, Results, first paragraph, last sentence):

„Likewise, although mean heart rate was not different (uninephrectomized mice: 460.7 bpm, normal controls: 462.4 bpm), uninephrectomized mice reached a lower miminal (336.5 bpm) and maximal (563.9 bpm) heart rate than controls (minimal heart rate:403.4 bpm, maximal heart rate: 567.6 bpm) under resting conditions.“

Old (Page 10, Discussion, second paragraph, first and second sentence):

As a negative finding, uninephrectomy alone was not associated with arterial hypertension in this study. This may relate to the relatively short observation time of 1 week. 

New (Page 10, Discussion, second paragraph, first sentence):

“As a negative finding, uninephrectomy alone was not associated with arterial hypertension in this study, even though the nominal reduction of resting minimal and maximal heart rate is consistent with an activation of baroreflex to buffer a small rise of blood pressure during the relatively short observation time of 1 week.”

---

## [Decision Letter · Decision Letter 2]

17 Feb 2022

PONE-D-21-32299R2Experimental uninephrectomy associates with less parasympathetic modulation of heart rate and facilitates sodium-dependent arterial hypertensionPLOS ONE

Dear Dr. Pliquett,

Thank you for submitting your manuscript to PLOS ONE. After careful consideration, we feel that it has merit but does not fully meet PLOS ONE’s publication criteria as it currently stands. Therefore, we invite you to submit a revised version of the manuscript that addresses the points raised during the review process.

Reviewer #2 is satisfied but reviewer #3 still has a major problem with the LF BPV data (see below). He is an expert in the field. Therefore this must be corrected, otherwise the manuscript cannot be accepted.In fact, he feels that besides correcting the data in the present manuscript an erratum should be published for the previous publication (Ref. 13).

We look forward to receiving your revised manuscript.

Kind regards,

Jaap A. Joles, DVM, PhD

Academic Editor

PLOS ONE

Reviewers' comments:

Reviewer's Responses to Questions

**Comments to the Author**

1. If the authors have adequately addressed your comments raised in a previous round of review and you feel that this manuscript is now acceptable for publication, you may indicate that here to bypass the “Comments to the Author” section, enter your conflict of interest statement in the “Confidential to Editor” section, and submit your "Accept" recommendation.

Reviewer #2: All comments have been addressed

Reviewer #3: (No Response)

2. Is the manuscript technically sound, and do the data support the conclusions?

Reviewer #2: Yes

Reviewer #3: No

3. Has the statistical analysis been performed appropriately and rigorously? 

Reviewer #2: Yes

Reviewer #3: Yes

4. Have the authors made all data underlying the findings in their manuscript fully available?

Reviewer #2: Yes

Reviewer #3: Yes

5. Is the manuscript presented in an intelligible fashion and written in standard English?

Reviewer #2: Yes

Reviewer #3: Yes

6. Review Comments to the Author

Reviewer #2: (No Response)

Reviewer #3: Except my previous comment #8, the authors have addressed all my previous comments appropriately.

However, the data for LF BPV shown in Figure 2 are wrong and not compatible with life. The animals would likely be dead if BPV was that high. A LF spectral power of BP of 80,000 mmHg2 is simply not possible !!!! The fact that the authors reported incorrect LF spectral powers in Fig. 4 of their previous publication (Ref 13), does not make these data correct. Again, my assumption is that the authors forgot to multiply the spectral powers with the spectral resolution, which was 0.019 Hz in this study. If the spectral powers reported in Fig. 2 of the current manuscript are multiplied with 0.019 Hz, the values would make sense. I.e., 80,000 mmHg2 (corresponding to BP amplitudes of 282 mmHg!) would become 1,520 mmHg2 (corresponding to BP amplitudes of 39 mmHg, which may be possible in hypertensive animals).

7. PLOS authors have the option to publish the peer review history of their article (what does this mean?). If published, this will include your full peer review and any attached files.

Reviewer #2: No

Reviewer #3: No

While revising your submission, please upload your figure files to the Preflight Analysis and Conversion Engine (PACE) digital diagnostic tool, https://pacev2.apexcovantage.com/. PACE helps ensure that figures meet PLOS requirements. To use PACE, you must first register as a user. Registration is free. Then, login and navigate to the UPLOAD tab, where you will find detailed instructions on how to use the tool. If you encounter any issues or have any questions when using PACE, please email PLOS at figures@plos.org. Please note that Supporting Information files do not need this step

---

## [Author Response · Author response to Decision Letter 2]

21 Feb 2022

Reviewer #3: 

We thank the reviewer for reevaluating our manuscript. Below, please find our reply.

Except my previous comment #8, the authors have addressed all my previous comments appropriately. However, the data for LF BPV shown in Figure 2 are wrong and not compatible with life. The animals would likely be dead if BPV was that high. A LF spectral power of BP of 80,000 mmHg2 is simply not possible !!!! The fact that the authors reported incorrect LF spectral powers in Fig. 4 of their previous publication (Ref 13), does not make these data correct. Again, my assumption is that the authors forgot to multiply the spectral powers with the spectral resolution, which was 0.019 Hz in this study. If the spectral powers reported in Fig. 2 of the current manuscript are multiplied with 0.019 Hz, the values would make sense. I.e., 80,000 mmHg2 (corresponding to BP amplitudes of 282 mmHg!) would become 1,520 mmHg2 (corresponding to BP amplitudes of 39 mmHg, which may be possible in hypertensive animals).

We thank you for this important comment. You are right, after reviewing the data analysis again, the multiplication of spectral powers with the spectral resolution is lacking. Now, we multiplied all spectral powers with the spectral resolution (0.019 Hz) both in this study and in the previous study cited as reference 13. There, we submit an erratum.

Figure 2 was changed, source data was updated.

Results (Page 9, last sentence) was updated:

Old: “As for the autonomic nervous system, parasympathetic modulation of heart rate (HF-HR) was not different between both hypertensive groups, however, sympathetic modulation (LF-SBP) was higher in the high-angiotensin-II model (105988 ± 6739 mmHg² versus 84424 ± 4604 mmHg², p=0.0038).”

New: “As for the autonomic nervous system, parasympathetic modulation of heart rate (HF-HR) was not different between both hypertensive groups, however, sympathetic modulation of vascular tone (LF-SBP) was higher in the high-angiotensin-II model (2014 ± 128.0 mmHg² versus 1604 ± 87.5 mmHg², p=0.0038).”

---

## [Decision Letter · Decision Letter 3]

23 Feb 2022

Experimental uninephrectomy associates with less parasympathetic modulation of heart rate and facilitates sodium-dependent arterial hypertension

PONE-D-21-32299R3

Dear Dr. Pliquett,

We’re pleased to inform you that your manuscript has been judged scientifically suitable for publication and will be formally accepted for publication once it meets all outstanding technical requirements.

Kind regards,

Jaap A. Joles, DVM, PhD

Academic Editor

PLOS ONE

Additional Editor Comments (optional):

Reviewers' comments:

Reviewer's Responses to Questions

**Comments to the Author**

1. If the authors have adequately addressed your comments raised in a previous round of review and you feel that this manuscript is now acceptable for publication, you may indicate that here to bypass the “Comments to the Author” section, enter your conflict of interest statement in the “Confidential to Editor” section, and submit your "Accept" recommendation.

Reviewer #3: All comments have been addressed

2. Is the manuscript technically sound, and do the data support the conclusions?

Reviewer #3: (No Response)

3. Has the statistical analysis been performed appropriately and rigorously? 

Reviewer #3: (No Response)

4. Have the authors made all data underlying the findings in their manuscript fully available?

Reviewer #3: (No Response)

5. Is the manuscript presented in an intelligible fashion and written in standard English?

Reviewer #3: (No Response)

6. Review Comments to the Author

Reviewer #3: The authors have addressed all remaining issues. I have no more comments or suggestions.

I think the journal of Ref. 13 is no longer existing. Not sure if an erratum is possible?

7. PLOS authors have the option to publish the peer review history of their article (what does this mean?). If published, this will include your full peer review and any attached files.

Reviewer #3: No

---

## [Editor Report · Acceptance letter]

28 Feb 2022

PONE-D-21-32299R3 

Experimental uninephrectomy associates with less parasympathetic modulation of heart rate and facilitates sodium-dependent arterial hypertension 

Dear Dr. Pliquett:

I'm pleased to inform you that your manuscript has been deemed suitable for publication in PLOS ONE. Congratulations! Your manuscript is now with our production department. 

Kind regards, 

on behalf of

Dr. Jaap A. Joles 

Academic Editor

PLOS ONE